# Detection and Genomic Characterization of Canine Circovirus in Iran

**DOI:** 10.3390/ani12040507

**Published:** 2022-02-17

**Authors:** Farzad Beikpour, Linda Amarachi Ndiana, Alireza Sazmand, Paolo Capozza, Farzad Nemati, Francesco Pellegrini, Salman Zafari, Seyed Masoud Zolhavarieh, Roberta Cardone, Reza Faraji, Gianvito Lanave, Vito Martella, Nicola Decaro

**Affiliations:** 1Department of Veterinary Medicine, University of Bari Aldo Moro, 70010 Valenzano, Italy; farzad.beikpour@uniba.it (F.B.); linda.ndiana@uniba.it (L.A.N.); paolo.capozza@uniba.it (P.C.); francesco.pellegrini@uniba.it (F.P.); roberta.cardone@uniba.it (R.C.); gianvito.lanave@uniba.it (G.L.); vito.martella@uniba.it (V.M.); 2Department of Pathobiology, Faculty of Veterinary Science, Bu-Ali Sina University, Hamedan 6517658978, Iran; alireza.sazmand@basu.ac.ir (A.S.); farzadnematidvm@gmail.com (F.N.); salman.zafari1376@gmail.com (S.Z.); 3Zoonotic Diseases Research Center, School of Public Health, Shahid Sadoughi University of Medical Sciences, Yazd 8915173160, Iran; 4Department of Clinical Sciences, Faculty of Veterinary Science, Bu-Ali Sina University, Hamedan 6517658978, Iran; mzolhavarieh@basu.ac.ir; 5Department of Animal Breeding & Genetics, College of Agriculture and Natural Resources, University of Tehran, Karaj 7787131587, Iran; reza.faraji@ut.ac.ir

**Keywords:** canine circovirus, non-diarrheic dogs, rectal swabs, Iran

## Abstract

**Simple Summary:**

During the last decade, canine circovirus (CaCV) has been repeatedly reported in domestic dogs with and without acute enteritis. Here we report the detection and full genome characterization of CaCV strains from non-diarrheic dogs in Iran. The results showed the circulation of the virus in this country, with Iranian strains segregating from the CaCVs detected in other countries.

**Abstract:**

Canine circovirus (CaCV) is a single-stranded DNA virus that globally circulates in dogs and wild carnivores. Although the pathogenic potential of the virus has not been fully understood yet, CaCV has been suggested to exacerbate the clinical course of other canine viral infections but also to circulate in dogs without clinical signs. In this study, we carried out real-time PCR assays to detect enteric pathogens from 156 canine rectal swabs collected from dogs without enteritis in 3 different regions in Iran. A total of 14 samples tested positive for CaCV and full-length genome sequences were obtained from 6 of the detected strains. Sequence and phylogenetic analyses showed that, despite the distance between the different sample collection sites, all Iranian CaCV strains were closely related and formed a separate clade from extant CaCVs. The present study shows that CaCV is circulating in non-diarrheic dogs in Iran, thus highlighting the need for further epidemiological investigations in Iranian domestic and wild carnivores.

## 1. Introduction

The family *Circoviridae* comprises viruses with covalently closed, circular, single-stranded DNA (ssDNA) genomes, including the smallest known autonomously replicating, capsid-encoding animal pathogens [1]. Circoviruses (CVs) are associated with fatal diseases in birds and pigs [2,3]. Canine circovirus (CaCV) was first detected in serum samples from dogs with vasculitis and/or hemorrhagic gastroenteritis (strain NY214, GenBank accession number JQ821392) [3]. The 2 kb-long genome contains two inversely arranged open reading frames (ORFs) coding for the rolling circle replication initiator protein gene (Rep) and the capsid protein gene (Cap) [1]. Subsequent studies in China [4], Thailand [5], Turkey [6], Italy [7], Germany [2], Brazil [8], Argentina [9], Vietnam [10] and Colombia [11] have revealed that CaCV is broadly circulating in domestic dog populations worldwide. Interestingly, CaCV and genetically related CVs have also been detected in other canids, including wolves and foxes [12,13,14,15,16]. Moreover, recent evidence shows the virus might have circulated in wild species for a long time before its first discovery in 2012 [16]. The ecology of CaCV in its animal hosts remains unexplored, and it is still unclear whether the virus can infect other animal hosts and what the modalities of transmission are between domestic and wild canids. Like other ssDNA viruses, CVs show a high genetic plasticity that might theoretically explain the ability of CaCV to cross the species barrier [13]. Although its pathogenic potential remains unclear, CaCV has been associated with respiratory and gastrointestinal disease and in some cases with systemic disease involving vasculitis [2,7,17,18,19]. Moreover, a possible synergism with other canine viral pathogens has been hypothesized [9,20]. In this paper, we investigated the presence of CaCV in dogs without diarrhea in three different provinces in Iran. The full-length genome sequences of 6 out of 14 detected strains were determined to gather information on the genetic heterogeneity of the CaCV strains.

## 2. Materials and Methods

### 2.1. Sample Collection

Between October 2018 and November 2019, a total of 156 rectal swabs were collected from dogs in 3 Iranian provinces in the south-west (Ahvaz; *n* = 71), west (Kermanshah; *n* = 67) and center (Yazd; *n* = 18) of the country. According to the Köppen–Geiger climate classification system [21], Ahvaz has a mid-latitude steppe and desert climate (Bsh) with an annual rainfall of 215 mm, Kermanshah has a warm and temperate climate (Csa) with the annual rainfall of 437 mm, while Yazd has a completely desert and dry climate (BWh) with about 55 mm precipitation annually (Figure 1).

Samples were collected from sheltered dogs. In Iran, shelters are mostly established and managed by non-governmental organizations (NGOs). Dogs in these shelters are rescued from all over the province. The dogs were clinically examined by a veterinarian, and animal data including age, sex, breed and neutering status were recorded for individuals. The cotton part of the sterile swab was put in plain 2 mL microtubes and transferred at a controlled temperature (<4 °C) to the Laboratory of Parasitology, Faculty of Veterinary Science, Bu-Ali Sina University, Iran, where they were stored at −20 °C before shipment to the Laboratory of Virology, Department of Veterinary Medicine, University of Bari Aldo Moro, Italy.

### 2.2. Nucleic Acid Extraction

Rectal swabs were homogenized in 1 mL of Dulbecco’s modified Eagle’s medium (DMEM) and then centrifuged at 10,000× *g* for 3 min. RNA/DNA extraction was performed, starting from two hundred microliters of the supernatants, with the QIAamp cador Pathogen Mini Kit (Qiagen S.p.A., Milan, Italy), following the manufacturer’s protocol, and stored at −80 °C until use.

### 2.3. Detection of Common Enteric Pathogens

DNA/RNA extracts were screened for enteric pathogens of dogs, including canine coronavirus [22,23], canine parvovirus [24,25], caliciviruses [26] and astroviruses [27]. Samples processed by real-time PCR were considered positive if the amplification curves exceeded the threshold line generated by the software on the basis of the background fluorescence. Samples processed by gel-based PCR were considered positive if 1.5% agarose-gel electrophoresis visualized amplicons of the expected size.

### 2.4. Molecular Detection of CaCV

A real-time PCR assay specific for CaCV [17] was used to process the nucleic acid extracts. The reaction was performed on a 7500 Real-time PCR System (Applied Biosystems, Foster City, CA, USA) using iTaq Supermix (Bio-Rad Laboratories Srl, Milan, Italy). The reaction mixture (25 µL) contained 12.5 µL of iTaq Supermix, 600 nmol/L of DogCV-forward and DogCV-reverse primers [17], 200 nmol/L of DogCV-probe [17] and 10 µL of template or plasmid DNA. The following thermal protocol was adopted: activation of iTaq DNA polymerase at 95 °C for 10 min and 45 cycles of denaturation at 95 °C for 15 s and annealing/extension at 60 °C for 1 min.

A nested PCR (nPCR) protocol with primers specific for CaCV [17] was performed to confirm the presence of the virus in real-time PCR positive samples. First-round PCR was performed using Platinum™ II Hot-Start Green PCR Master Mix (2×) (Invitrogen™ Thermo Fisher Scientific, Shanghai, China). The thermal protocol included a first denaturation step at 94 °C for 2 min, followed by 35 cycles of 94 °C for 15 s, 52 °C for 15 s and 68 °C for 15 s. One microliter of a 1:100 dilution of the PCR product was added in a second-round PCR using the same mix and thermal protocol used for the first-round PCR. To avoid cross-contamination, a strict separation between pre-amplification for reaction mix preparation, sample preparation and addition to the mix and amplification/post-amplification was adopted. The DNA extracts were added to the mixtures in separate laminar flow cabinets. Homogenized liver specimens from dogs with CV infections were used as positive controls, while distilled water was used as negative control.

Contaminations were avoided through the application of rigorous laboratory procedures and good laboratory practices (i.e., handling samples separately from positive controls and using negative controls and extraction controls during extraction and real-time/conventional PCR). The amplicons with the correct size were purified using specific enzymes (Exo1 and FastaP) or gel purified by a Qiaquick PCR Purification Kit (Qiagen GmbH, Hilden, Germany) to remove primer dimers and/or aspecific bands (incorrect size). The purified PCR products with sufficient DNA concentrations (>10 ng/μL) were subsequently sent to an external laboratory by Eurofins Genomics GmbH (Ebersberg, Germany) to perform Sanger sequencing. The obtained Sanger sequences were subjected to sequence analyses using Geneious software version 10.2.4 (Biomatters, Auckland, New Zealand). High-quality sequences (>95%) were subjected to BLAST N and FASTA Nucleotide searches in order to retrieve the best hit in the NCBI and EBI sequence databases, respectively.

### 2.5. Strategy for Full-Genome Amplification of CaCVs

On the basis of the direct sequencing results, CaCV-positive samples were selected to perform amplification of the full-length CaCV genome. To increase the number of circular genomes in the samples, a protocol based on the rolling circle amplification (RCA) technique [28] was performed using the TempliPhi 100 amplification kit (GE Healthcare, Milan, Italy), with minor modifications. Briefly, 5 µL of extracted DNA were added to a mix containing 4 µL of TempliPhi sample buffer and 1 µL of universal circovirus primer reverse [29]. Then, the mixture was incubated at 95 °C for 3 min and subsequently cooled on ice, and 5 µL TempliPhi reaction buffer supplemented with 0.7 µL of 10mM dNTPs and 0.2 µL TempliPhi enzyme was added to the annealing mix. Finally, the mixture was incubated at 35 °C for 22 h and subsequently inactivated at 65 °C for 15 min.

In order to obtain the complete genome of DogCVs, a back-to-back (BTB) PCR approach was performed on the RCA products using forward BTB DOG CV 446 F (WCTCGCGAGGSTTGCGAGASCT) and reverse BTB DOG CV 273 R (HCCCCIAGCAGGCTCAAAITGKCC). Briefly, BTB PCR was performed in a final volume of 50 μL containing 5 μL of RCA product and TaKaRa LA Taq^TM^ (Takara Bio Europe S.A.S., Saint-Germain-en-Laye, France) mix consisting of 19.5 μL of PCR-grade water, 5 μL of 10× buffer, 5 μL of MgCl_2_ (25 mM), 1 μL of the forward and reverse primers (50 μM), 8 μL of deoxynucleotide triphosphates (dNTPs) (2.5 mM) and 0.5 μL of Takara La Taq polymerase (5 U/μL). The PCR was run as follows: initial denaturation at 94 °C for 2 min, followed by 35 cycles of denaturation at 94 °C for 30 s, annealing at 60 °C for 30 s and extension at 68 °C for 3 min, followed by a final extension at 68 °C for 10 min. Subsequently, a nested PCR approach was performed on a 1:100 dilution of BTB PCR products using forward BTB DOG CV 548 F (GCAAGAGCCGGTAYTGCATGGA) and reverse BTB DOG CV 165 R (YTCCCCIACCTCCCGRCCACARAT) primers and TaKaRa LA TaqTM Kit (Takara Bio Europe S.A.S., Saint-Germain-en-Laye, France) mix. The same thermal conditions used for the BTB PCR were also applied to the nested PCR. The primers used for the BTB and nested PCR were designed based on the CaCV genome sequences retrieved from the NCBI database (htttp://www.ncbi.nlm.nih.gov, accessed on 12 February 2019) using the Primer3 plugin implemented in Geneious software version 10.2.4 (Biomatters, Auckland, New Zealand). The PCR products were analyzed on a 1.5% agarose gel prepared in TBE buffer (0.09 M of boric acid, 0.09 M of Tris and 0.0025 M of EDTA, pH 8.3) and subjected to electrophoresis at 50 V for 90 min. The amplification bands were visualized on a Gel Doc™ EZ (Bio-Rad Laboratories SRL, Segrate, Italy), purified and subjected to direct Sanger sequencing in both directions.

### 2.6. Sequence and Phylogenetic Analyses

The obtained genomic sequences of the CaCV strains detected in this study were assembled and analyzed using the Geneious software package version 10.2.4 (http://www.geneious.com, accessed on 8 January 2021) and the online tools BLAST (https://blast.ncbi.nlm.nih.gov/blast.cgi, accessed on 8 January 2021) and FASTA (https://www.ebi.ac.uk/tools/sss/fasta/nucleotide.html, accessed on 8 January 2021) with default values to find homologous hits.

The appropriate substitution model settings for the phylogenetic analysis were derived using “Find the best DNA/Protein models” implemented in MEGAX software (https://www.megasoftware.net; accessed on 10 January 2021) [30,31]. The phylogenetic analyses were performed with MEGAX software using the maximum-likelihood method with the general time-reversible model, a proportion of invariant sites and a discrete gamma distribution (6 categories) to model evolutionary rate differences among sites and bootstrap analyses with 1000 pseudoreplicated datasets. The obtained sequences were aligned with cognate circovirus strains retrieved from the GenBank database by the MAFFT algorithm [31].

Phylogenetic analyses using other evolutionary models (Bayesian inference and neighbor joining) were performed to compare the topology of the phylogenetic trees. Similar topologies with slight differences in bootstrap values at the nodes of the tree were observed. Accordingly, the maximum-likelihood tree was retained.

## 3. Results

Out of 156 examined dogs, 14 (8.9% prevalence; 95% confidence interval 5–14.6) tested positive for CaCV by real-time PCR assay. Their cycle threshold (C*_T_*) values were generally low, ranging from 20.75 to 35.53. Infection rates were highest in Yazd (5/18; 27.8%) followed by Ahvaz (5/71; 7.04%) and Kermanshah (4/67; 5.9%) provinces. The average age of infected dogs was 3 years and 4 months (95% CI, 9 months–5 years and 11 months), which is higher than that of studied population, i.e., 2 years and 9 months (95% CI, 2 years, and 6 months–3 years and 3 months). The demographic and clinical data of the 14 infected dogs are shown in Table 1, which shows that CaCV infected dogs had no enteric signs. Other tested pathogens i.e., coronavirus, parvovirus, caliciviruses and astroviruses were not detected.

The complete CaCV genome sequence was obtained only from six Iranian dogs, likely due to the low viral DNA content of most samples. The genomes of the 6 CaCVs sequenced in this study was 2063 nt in length, like all other CaCVs. The genome features included two major open reading frames (ORFs), located on complementary strands in the opposite orientation. These were ORF1 (912 nt) on the virion strand and ORF2 (813 nt) on the complementary strand of the replicative form encode for the Rep (303 amino acids) and Cap (270 amino acids) proteins, respectively. Similar to other animal CVs, the genome of the Iranian strains contained two intergenic non-coding regions of 135 and 203 nt in length, which were located between the start and stop codons of the replicase and capsid protein genes, respectively. The 5′-intergenic region contained a thermodynamically stable stem-loop, which regulates the initiation of rolling-circle replication, and the conserved mononucleotide motif TAGTATTAC [3,17].

The nucleotide alignment between the complete genomic sequences of CaCVs obtained in this study and the reference strains retrieved from the GenBank database displayed an overall nucleotide (nt) identity ranging from 80.8% to 100%. All the Iranian CaCV strains were distinguishable in one group with an overall genome nucleotide identity ranging from 92.0 to 98.7% to each other. Strains IRN/2019/Dog/292, IRN/2019/Dog/298, IRN/2019/Dog/308 IRN/2019/Dog/298 and IRN/2019/Dog/497 shared the highest nt identity (92.5–92.8%) to CaCV strain AZ2972-13, which was detected in a dog in Italy in 2013 (GenBank accession nr KT734813), while strains IRN/2019/Dog/496 and IRN/2019/Dog/498 shared the highest nt identity (93.0–93.3%) to CaCV strain VN-8, which was detected in dog in Vietnam in 2018 (GenBank accession nr MT740200).

The unrooted phylogenetic tree based on the complete genome nucleotide sequences (Figure 2) showed a well distinguishable clustering of the CaCV strains into six groups with group 1 including CaCVs identified in dogs, wolves and a badger from Europe, America and Asia, groups 2, 3 and 4 comprising CaCVs discovered in dogs from Asia and a red fox in Italy and group 5 containing CaCVs identified in artic and red foxes in different European countries. The sequences detected in Iranian dogs belong to a separate clade, tentatively numbered as group 6 (Figure 2). The nt identity values calculated for the CaCV sequences belonging to the same cluster and between different clusters identified in the phylogenetic analysis are reported in Table 2.

The deduced amino acid sequence of the ORF1 and ORF2 genes of the Iranian CaCV strains retrieved in this study and a selection of CaCV strains belonging to the phylogenetic groups (Table 2) were compared (see Appendix A). The comparison in the ORF2 region showed 10 unique non-synonymous mutations in the Iranian CaCVs as compared to CaCVs belonging to other phylogenetic groups. In detail, unique ORF2 amino acid variations were found in 4 putative genotype-specific positions [32]: 13 (threonine residue in Iranian strains #308, 496 and 498), 50 (methionine residue in Iranian strain #498), 148 (arginine residue in all Iranian strains) and 178 (glutamine residue in Iranian strain #308) (Appendix A). Further unique ORF2 amino acid variations were found in 6 other positions: 23 (lysine residue in Iranian strain #298), 83 (isoleucine residue in Iranian strain #298), 103 (lysine residue in Iranian strain #498), 123 (isoleucine residue in Iranian strain #308), 231 (isoleucine residue in Iranian strain #497) and 234 (leucine residue in Iranian strain #497) (Appendix A).

The comparison in the ORF1 region showed 22 unique non-synonymous mutations in the Iranian CaCVs as compared to strains belonging to other phylogenetic groups.

In detail, unique ORF1 amino acid variations were found consistently in more than 1 Iranian strain in 7 positions: 8 (lysine residue in Iranian strains #298, 496 and 498), 10 (alanine residue in Iranian strains #298, 496 and 498), 32 (threonine residue in Iranian strains #292, 298, 308, and 497), 69 (glutamine residue in all Iranian strains), 174 (glutamine residue in Iranian strains #496, 497 and 498), 282 (serine residue in Iranian strains #298, 496 and 498) and 300 (leucine residue in Iranian strains #298, 496 and 498) (Appendix A).

### GenBank Sequence Submission

Nucleotide sequences of strains IRN/2019/Dog/292, IRN/2019/Dog/298, IRN/2019/Dog/308, IRN/2019/Dog/496, IRN/2019/Dog/497 and IRN/2019/Dog/498 used for phylogeny were deposited in the GenBank under accession nos. OK625288, OK625289, OK625290, OK625291, OK625292 and OK625293, respectively.

## 4. Discussion

In the present study, we report the preliminary data on the detection and full-genome characterization of CaCVs in swab samples collected from non-diarrheic dogs in Iran. Circulation of CaCV in different countries worldwide has been consistently reported [2,4,5,6,7,8,9,10,11,16]. However, no report of CaCV detection has been described yet in Iran, thus limiting the current understanding about the circulation of CaCV within the country. A CaCV infection rate of 6% has been reported in western neighboring Turkey [6], mirroring the prevalence of 9% retrieved in this study. Interestingly, almost two-thirds (64%) of the CaCV-positive Iranian samples originated from western provinces of the country, geographically close to Turkey, thus allowing us to hypothesize a possible introduction of the virus from neighboring countries or an underestimation of the presence of the CaCV in Iran.

The dogs sampled in this study were in good health conditions or displayed clinical signs not involving the gastroenteric tract. No other enteric viral pathogens were detected in the samples analyzed. Since its first identification, there is no agreement in the scientific community about the pathogenetic role of CaCV. Indeed, CaCV has been detected repeatedly in samples (stools, serum and nasal swabs) both from healthy animals and from dogs with respiratory and gastrointestinal disease and vasculitis [2,7,17,18,19]. While previous reports accounted for a primary role in the development of fatal hemorrhagic enteritis in puppies [7] and vasculitis in dogs [17], subsequent studies have reported CaCVs in association with other pathogens, including canine parvovirus, canine enteric coronavirus, canine influenza virus, canine parainfluenza virus and canine respiratory coronavirus, in feces and respiratory samples, suggesting that the occurrence of coinfections may increase the severity of the clinical disease and the fatality rates [2,4,5,7,12,17,18,20].

We detected CaCV more frequently in adult dogs (9 months–20 years) (Table 1), which disagreed with the previous findings [7,12,20]. The age-related decrease in the prevalence of CaCV, which is usually observed [7,12,20], may be due to the development of a specific immune response induced by previous infections with a progressive clearance of the infectious agent. Therefore, this immune situation could account for a possible role of adult animals as asymptomatic CaCV carriers.

In our study, no wild carnivores i.e., foxes and wolves were sampled, thus preventing the evaluation of the epidemiological role of these animals in Iran. To date, circulation of CaCV in wild carnivores has been described only in European countries [12,13,14,16].

From the complete genome alignment carried out in this study, all the strain sequences identified in Iranian dogs showed a ≥80.8% nt identity to other CaCV retrieved from the GenBank. According to the species demarcation threshold of 80% for genome-wide nucleotide sequence identity for members of the family *Circoviridae* [1], the strains detected in this study belong to the CaCV species.

The phylogeny provided evidence of cluster formation of the CaCV sequences into six groups, except for CaCV UCD3-478 [17], which could represent an intermediate sequence between groups 4 and 5 [16]. Four groups correspond to genotypes 1 to 4 [4], while a fifth group is composed only by CaCVs identified in foxes [16], suggesting a general clustering on a geographic and/or host basis [4]. The Iranian strains analyzed in this study showed a well-defined cluster with a range of nt identity within and between the groups consistent with other clusters previously identified. By comparison of the deduced ORF1 and ORF2 sequences of the Iranian strains with selected CaCVs from other phylogenetic clusters, several unique non-synonymous mutations were observed. To date, nine major variable regions have been identified within the cap gene of CaCV [33]. Four mutations present in the cap gene of the Iranian strains is encompassed at these proposed locations (residues 13, 50, 148 and 178), suggesting that, until sequences from other geographical areas are analyzed, these variable regions need to be fully elucidated. Point mutations in the cap gene have been previously observed for porcine circovirus type 2 (PCV2) and are a potential source of virus evolution, especially due to selective pressure caused by vaccination [34]. In the case of CaCV, where vaccination has not been used, the mechanisms behind these aa substitutions are unclear; virus mutations under natural infection have been demonstrated to be very complex for PCV-2 [34]. Moreover, the importance of these changes is difficult to be elucidated, as the role of CaCV as a sole cause of disease is still under investigation.

## 5. Conclusions

The detection of CaCV strains in dogs in different provinces of Iran accounts for its widespread circulation in this country and highlights the need for extensive epidemiological surveillance in domestic and wild carnivores.

## Figures and Tables

**Figure 1 animals-12-00507-f001:**
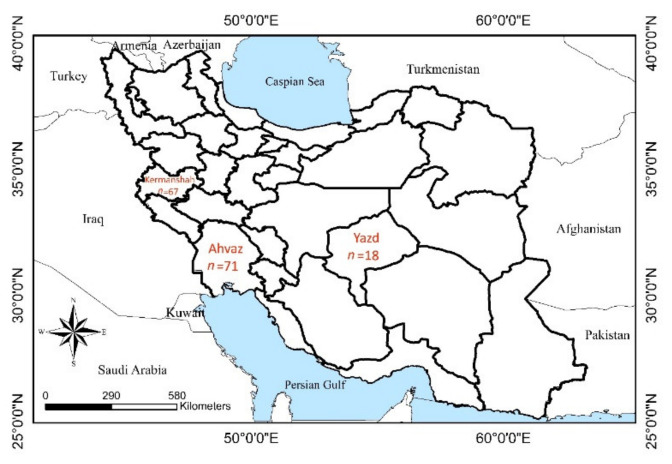
Sampled localities in Iran. Three different provinces were selected with different climate conditions, and in each city the main shelter where dogs are rescued from all over the province was sampled.

**Figure 2 animals-12-00507-f002:**
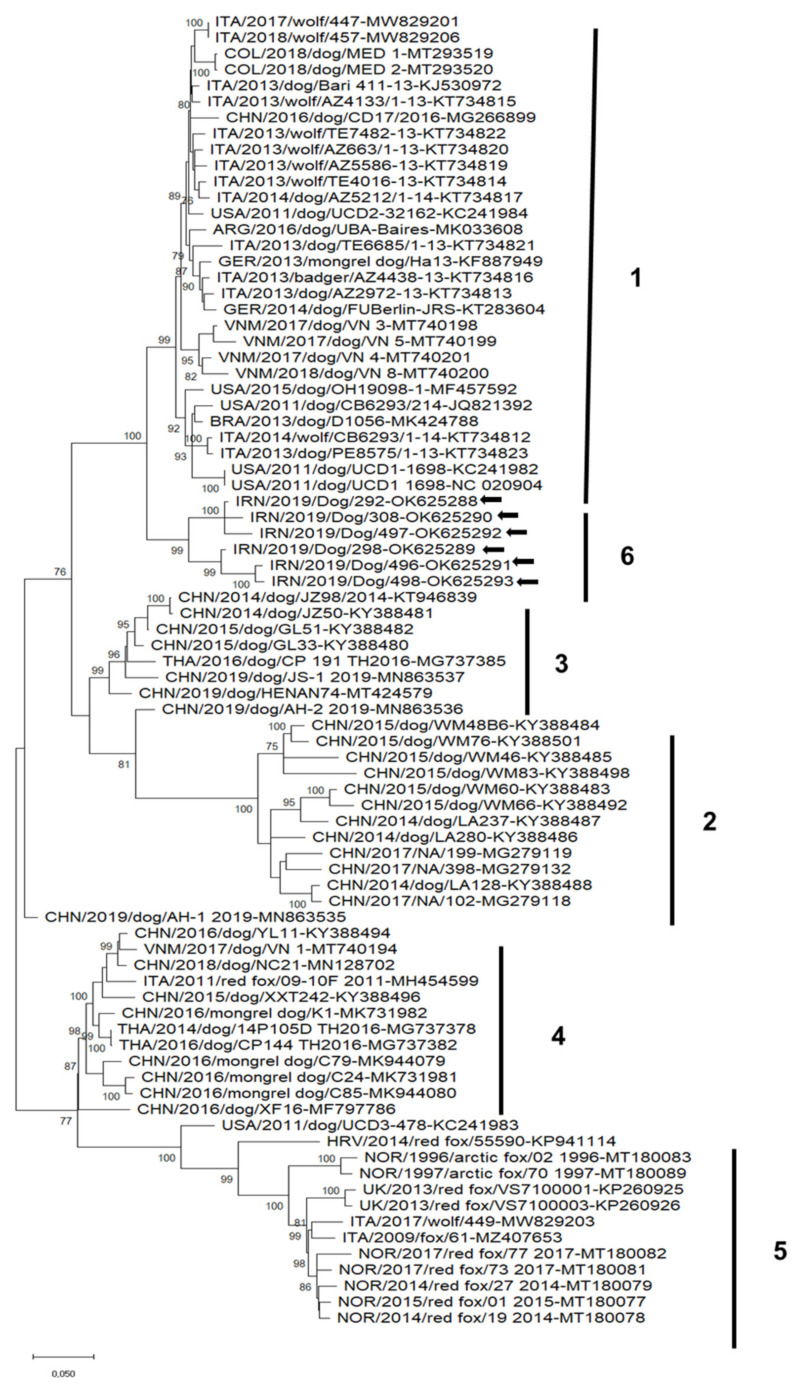
Unrooted phylogenetic tree constructed on the complete genome nucleotide sequences of canine circovirus strains obtained in this study and reference strains retrieved in the GenBank database. This phylogeny was carried out using the maximum-likelihood method and general time-reversible (GTR) model with a gamma distribution and invariable sites. The robustness of the individual nodes on the phylogenetic tree was estimated using 1000 bootstrap replicates. Bootstrap values greater than 75% are indicated. Black arrows indicate the strains detected in this study. The scale bars indicate the estimated number of nucleotide substitutions. Numbers 1-6 indicate the phylogenetic groups.

**Table 1 animals-12-00507-t001:** Signalment of the 14 Iranian dogs with canine circovirus infection.

#	Province	Sample Code	CaCV Real-Time PCR (C*_T_* Values)	Age	Sex	Neutered	General Health Condition	Clinical Abnormalities
1	Kermanshah	IRN/2019/Dog/292	20.75	1 y	F	No	Good	Moist mucous membranes
2	Kermanshah	IRN/2019/Dog/298	21.44	1 y	F	No	Good	None
3	Kermanshah	IRN/2019/Dog/300	22.59	1.5 y	M	No	Good	Tick infestation
4	Kermanshah	IRN/2019/Dog/308	21.94	5 y	F	No	Good	Moist mucous membranes
5	Yazd	IRN/2019/Dog/485	34.81	3 y	F	No	Good	Moist mucous membranes
6	Yazd	IRN/2019/Dog/486	26.94	20 y	M	No	Good	Moist mucous membranes
7	Yazd	IRN/2019/Dog/496	29.90	1 y	M	Yes	Good	Moist mucous membranes
8	Yazd	IRN/2019/Dog/497	35.53	2 y	F	No	Good	Moist mucous membranes
9	Yazd	IRN/2019/Dog/498	27.15	2.5 y	M	No	Good	Moist mucus membranes
10	Ahvaz	IRN/2019/Dog/615	33.98	1.5 y	F	Yes	Good	None
11	Ahvaz	IRN/2019/Dog/616	29.73	3.5 y	M	Yes	Fair	Serous nasal discharge
12	Ahvaz	IRN/2019/Dog/618	29.96	2 y	F	Yes	Fair	None
13	Ahvaz	IRN/2019/Dog/619	29.73	2 y	F	No	Poor	None
14	Ahvaz	IRN/2019/Dog/644	29.96	9 m	F	No	Poor	Serous nasal discharge, alopecia and swollen submandibular lymph node

y, years; m, months; F, female; M, male.

**Table 2 animals-12-00507-t002:** Nucleotide identities (%) between canine circovirus phylogenetic groups based on the overall genome (modified from Urbani et al., 2021).

	Group 1	Group 2	Group 3	Group 4	Group 5	UCD3-487	Group 6
Group 1	93.5–100	83.8–87.3	88.7–90.4	85.9–92.6	82.5–84.9	82.9–84.9	90.9–91.8
Group 2	83.8–87.3	89.1–99.9	85.1–90.1	84.8–90.6	80.8–90.6	81.9–84.6	84.3–89.2
Group	88.7–90.4	85.1–90.1	96.4–99.8	85.9–89.8	82.1–83.8	84.3–84.5	88.7–90.1
Group 4	85.9–92.6	84.8–90.6	85.9–89.8	92.4–99.9	84.4–86.6	88.8–90.9	85.6–89.4
Group 5	82.5–84.9	80.8–83.6	82.1–83.8	84.4–86.6	91.9–99.8	88.8–89.9	83.2–84.5
Group 6	90.9–91.8	84.3–89.2	88.7–90.1	85.6–89.4	83.2–84.5	84.4–85.3	92.0–98.7
UCD3-478	82.9–84.9	81.9–84.6	84.3–84.5	88.8–90.9	88.8–89.9	100	84.4–85.3

## Data Availability

The data presented in this study are available in this manuscript. Sequence data presented in this study are openly available in the GenBank database.

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
