# Peer review of "Detection and Genomic Characterization of Canine Circovirus in Iran"

_animals, 2022, doi:10.3390/ani12040507_

Round 1

Reviewer 1 Report

The paper by Beikpour Farzad present a group of interesting results about the circulation of Canine Circovirus in healthy dogs from Iran. The paper has strong phylogenetic support. This important result should be better discussed for a better publication.

Discussion section is too short. The role of CaCV as a single non pathogenic agent must be discussed. Also It would be desirable to enhance the discussion about the role of CaCV as a pathogen alone or in Co-Infection to other pathogens, as has been reported in Brasil where CaCV has been reported in Healthy (10.1371/journal.pone.0218735 ) and Enteric diseased Dog samples (https://doi.org/10.1590/0103-8478cr20190909).

Figure 2 has too low resolution; Branch support cannot be seen. Tables must be unified on its presentation (Without vertical lines).

Please Include new references from CaCV circulation in Asia -Vietnam (doi: 10.1080/01652176.2021.1967511 ) and South America -Colombia (DOI: 10.1038/s41598-020-74630-8 ) 

Author Response

Comments from Reviewer 1 (R.1)

The paper by Beikpour Farzad present a group of interesting results about the circulation of Canine Circovirus in healthy dogs from Iran. The paper has strong phylogenetic support. This important result should be better discussed for a better publication.

R.1.1: Discussion section is too short. The role of CaCV as a single nonpathogenic agent must be discussed. Also, it would be desirable to enhance the discussion about the role of CaCV as a pathogen alone or in Co-Infection to other pathogens, as has been reported in Brasil where CaCV has been reported in Healthy (10.1371/journal.pone.0218735) and Enteric diseased Dog samples (https://doi.org/10.1590/0103-8478cr20190909).

Reply to R1.1: We thank the referee for the suggestion. Accordingly, we implemented information in the discussion section regarding the pathogenic role of CaCV in dogs as a pathogen alone or in co-infection to other pathogens (lines 650-665).

R.1.2: Figure 2 has too low resolution; Branch support cannot be seen. Tables must be unified on its presentation (Without vertical lines).

Reply to R1.2: We agree with the observation of Reviewer 1.  Following the suggestion of the referee, we have implemented the resolution of Figure 2 and we unified the tables in the revised manuscript (Figure 2 and Tables)

R.1.3: Please Include new references from CaCV circulation in Asia -Vietnam (doi: 10.1080/01652176.2021.1967511) and South America -Colombia (DOI: 10.1038/s41598-020-74630-8).

Reply to R1.3: Following the suggestion of the Reviewer 1 we have included the references from CaCV circulation in Asia -Vietnam (doi: 10.1080/01652176.2021.1967511) and South America -Colombia (DOI: 10.1038/s41598-020-74630-8) (see refs 10 and 11 in text and reference list). 

Reviewer 2 Report

In this paper, authors investigated the prevalence and genetic characterization of canine circovirus in domestic dogs without clinical signs in three Iranian provinces.  They revealed circulating CaCVs in Iran and their genetic characteristic. The aim of this study is clear and the study design is straightforward. Unfortunately, this paper can not provide more useful information for the study of epidemiology and pathogenicity of canine circovirus. The manuscript in the present format cannot be accepted for publication in this journal. The reviewer has some generalized and specific comments for the authors to address.

  1. Authors only investigated the circulation of CaCVs in domestic dogs without clinical signs. Although the pathogenic potential of CaCVs has not been fully understood, CaCV has been suggested to exacerbate the clinical course of other canine viral infections. Why the investigation is not carried out in dogs with clinical signs. The investigation results may be useful for the understand of the pathogenicity of canine circovirus.
  2. Out of 156 rectal swabs, 14 tested positive for CaCV. However, only 6 samples were selected for the sequencing of full-length genome. Why?
  3. Authors collected rectal samples from examined dogs for the detection of nucleic acid of CaCVs. CaCVs also may be present in the upper respiratory tract, the investigation of CaCVs in nasal swabs can provide more evidence for the circulation of CaCVs.
  4. Sequences analysis was carried out with the purified amplification bands. How did the authors exclude the possibility of PCR errors and contamination?
  5. Phylogenetic analysis is really rough and should be improved.
  6. 2 should be clearer.
  7. Line 54, please supplement the GenBank accession number of strain NY214.

Author Response

Comments from Reviewer 2 (R.2)

In this paper, authors investigated the prevalence and genetic characterization of canine circovirus in domestic dogs without clinical signs in three Iranian provinces.  They revealed circulating CaCVs in Iran and their genetic characteristic. The aim of this study is clear and the study design is straightforward. Unfortunately, this paper can not provide more useful information for the study of epidemiology and pathogenicity of canine circovirus. The manuscript in the present format cannot be accepted for publication in this journal. The reviewer has some generalized and specific comments for the authors to address.

R.2.1: Authors only investigated the circulation of CaCVs in domestic dogs without clinical signs. Although the pathogenic potential of CaCVs has not been fully understood, CaCV has been suggested to exacerbate the clinical course of other canine viral infections. Why the investigation is not carried out in dogs with clinical signs. The investigation results may be useful for the understand of the pathogenicity of canine circovirus.

Reply to R2.1: The samples were collected randomly to evaluate the circulation of the virus within the canine population in different Iranian regions and to characterize the detected strains at the  molecular level, in order to assess the evolutionary relationship with CaCVs detected in other parts of the world. We did not aim to evaluate the association of CaCVs to specific clinical signs. We agree with the reviewer that further epidemiological studies will be needed to better understand the role of the virus in the occurrence of canine enteritis.

R.2.2: Out of 156 rectal swabs, 14 tested positive for CaCV. However, only 6 samples were selected for the sequencing of full-length genome. Why?

Reply to R2.2: In this study, sequences of about 2,063 nt in length were obtained from 6 of 14 samples; this was likely due to the fact that most samples contained only low amounts of CaCV DNA that prevented the amplification of a large genomic region. This was added to lines 540-541.

R.2.3: Authors collected rectal samples from examined dogs for the detection of nucleic acid of CaCVs. CaCVs also may be present in the upper respiratory tract, the investigation of CaCVs in nasal swabs can provide more evidence for the circulation of CaCVs.

Reply to R2.3: We agree with Reviewer 2 that CaCVs can be also associated with mild respiratory infection and episodic enteritis, but the aim of this study was to show data regarding the prevalence of CaCVs in rectal samples randomly collected from the canine population. The absence of respiratory signs in the tested population suggested us to test only fecal samples, since the intestinal mucosa has been identified as the primary site of replication for CaCV.  In subsequent epidemiological studies we will also include upper respiratory tract samples in addition to rectal samples as suggested by reviewer 2, especially in dogs with respiratory signs.

R.2.4: Sequences analysis was carried out with the purified amplification bands. How did the authors exclude the possibility of PCR errors and contamination?

Reply to R2.4: the samples were initially screened using a highly sensitive and specific real-time PCR, which is close system that prevents the carry-over contamination. Only the real-time PCR positive samples were submitted to conventional PCR with primers specific for CaCVs in order to obtain fragments useful for sequencing. Contaminations were avoided through the application of rigorous laboratory procedures that involve handling samples separately from positive controls and using negative controls and blanks during each step.

R.2.5: Phylogenetic analysis is really rough and should be improved.

Reply to R2.5: We agree with the referee and accordingly, we added more information in the text and we used more complex phylogenetic methods to confirm the obtained results (lines 300-372).

R.2.6: 2 should be clearer.

Reply to R2.6: Resolution of Figure 2 was implemented in order to clarify the phylogeny of the circovirus sequences retrieved in this study and the bootstrap support on the branches of the phylogenetic tree. The description of the phylogeny is detailed in the results section.

R.2.7: Line 54, please supplement the GenBank accession number of strain NY214.

Reply to R2.7: Following the suggestion of the Reviewer 2 we have provided the accession number (lines 54-55).

Reviewer 3 Report

Comments for the authors

The manuscript by Beikpour Farzad et al., has performed a very important study, and they have identified and characterized the full genome of CaCV strains from non-diarrheic dogs in Iran. The manuscript is well written however I have few comments for the authors.

  1. Are there any reported transmission cases of CaCV strains from dogs to humans?
  2. Is there any difference in the protein sequences of the CaCV strains observed in the diarrheic versus non-diarrheic dogs?

  1. What is the main question addressed by the research?
  • The manuscript by Beikpour Farzad et al., has performed a very important study. In this work, authors have collected rectal samples non-diarrheic dogs in Iran and identified and characterized the full genome of Canine circovirus (CaCV) strains.

  1. Do you consider the topic original or relevant in the field, and if
    so, why?
  • This study is relevant to the Canine virus biology as it improves our knowledge and understanding.

  1. What does it add to the subject area compared with other published
    material?
  • Except the new CaCV strains, it does not any new information to the subject area

  1. What specific improvements could the authors consider regarding the
    methodology?
  • Authors have used only PCR based method for sequencing the genomes. It is sufficient

  1. Are the conclusions consistent with the evidence and arguments
    presented and do they address the main question posed?

-Yes

  1. Are the references appropriate?

- Yes

  1. Please include any additional comments on the tables and figures.
  2. Authors have identified the viral strains from the dogs that do not have clinical signs. Are these strains can be considered as non-pathogenic? If not, what’s the significance of identification?
  1. Is there any difference in the protein sequences of the CaCV strains observed in the diarrheic versus non-diarrheic dogs?
  2. Are there any reported transmission cases of CaCV strains from dogs to humans?
  3. Figure 2 can be improved to make it more legible.

Author Response

Comments from Reviewer 3 (R.3)

The manuscript by Beikpour Farzad et al., has performed a very important study, and they have identified and characterized the full genome of CaCV strains from non-diarrheic dogs in Iran. The manuscript is well written however I have few comments for the authors.

R3.1: Are there any reported transmission cases of CaCV strains from dogs to humans?

Reply to R3.1: To date, since the first discovery of CaCV in 2012 by Kapoor et al. there is no evidence that canine circovirus can be transmitted to humans or cause human disease. Evidence for circovirus infection in mammals other than pigs is equivocal, and studies have been restricted to avian and porcine circoviruses (PCVs). Viral, circular DNA genomes have been detected in human stool samples and genetically characterized as belonging to the family Circoviridae. Circoviridae detection in the stools of healthy human adults was limited to porcine circoviruses which were also found in most pork products thus reflecting dietary consumption of PCV-infected pork. Moreover, reports regarding serological analysis have provided evidence that PCV is not infectious for humans.

R3.2: Is there any difference in the protein sequences of the CaCV strains observed in the diarrheic versus non-diarrheic dogs?

Reply to R3.2: In this study CaCV infected dogs had no enteric signs and the requested comparison diarrheic versus non-diarrheic dogs could not be done. In order to cope with the referee’ request we compared replicase and capsid proteins of the six Iranian strains with CaCV sequences selected from phylogenetic groups from Table 2. The CaCVs sequences from literature were also selected chiefly taking advantage of sequences of CaCVs identified from diarrheic dogs. We added supplementary Tables 1 and 2 to provide information of mutations in replicase and capsid proteins. Also, information regarding nt and aa mutations in replicase and capsid proteins was inserted in the results section (lines 575-597) and discussed in the manuscript (lines 822-835).

Reviewer 4 Report

The study investigated the genetic characters of prevalence of Canine circovirus (CaCV) in Iran. The results confirmed that CaCV can be transmitted in non-diarrheal dogs, and all the Iranian CaCV strains in the investigation were closely related and formed a separate clade from extant CaCVs. These results contribute to the prevention and control of CaCV in the future. The manuscript would be considered to accept for publication after making the minor revisions as bellows:

Q1:The reason for selecting sample areas based on climate was not explained. Is there any research to prove that climate play any roles on the epidemic of circovirus?

Q2:Please explain the details of the parentheses in line 181 "14(8.9%; 95% CI, 5-14.6) tested positive"

Q3: A total of 14 positive samples were detected. Why did the genome sequences of those CaCV only be selected from six Iranian dogs?

Q4: Taiwan is an inalienable part of China’s territory. I really think the English words expression ''Taiwan [16], and China [4]'' in line 259 is a mistake and should be corrected to Taiwan of P.R. China.

Author Response

Comments from Reviewer 4 (R.4)

The study investigated the genetic characters of prevalence of Canine circovirus (CaCV) in Iran. The results confirmed that CaCV can be transmitted in non-diarrheal dogs, and all the Iranian CaCV strains in the investigation were closely related and formed a separate clade from extant CaCVs. These results contribute to the prevention and control of CaCV in the future. The manuscript would be considered to accept for publication after making the minor revisions as bellows:

R4.1: The reason for selecting sample areas based on climate was not explained. Is there any research to prove that climate play any roles on the epidemic of circovirus?

Reply to R4.1: The collection of samples was performed in 3 different areas of South-West (Ahvaz), West (Kermanshah) and Center (Yazd) Iran provinces in order to guarantee a wide sampling throughout the country. Based on the present knowledge, there is no association between CaCV prevalence and climate. The reason for providing geographic and climate details is mostly due to provide a detailed description of the sampling areas that are more probably unknown to the readers of the Journal.

R4.2: Please explain the details of the parentheses in line 181 "14 (8.9%; 95% CI, 5-14.6) tested positive"

Reply to R4.2: We have now explain in the text (line 376). The confidence interval, set at 95%, expresses the statistical margin of error (α=0.05).

R4.3: A total of 14 positive samples were detected. Why did the genome sequences of those CaCV only be selected from six Iranian dogs?

Reply to R4.3: In this study, sequences of about 2,063 nt in length were obtained from 6 of 14 samples; this was likely due to the fact that most samples contained only low amounts of CaCV DNA that prevented the amplification of a large genomic region. This was added to lines 540-541.

R4.4: Taiwan is an inalienable part of China’s territory. I really think the English words expression ''Taiwan [16], and China [4]'' in line 259 is a mistake and should be corrected to Taiwan of P.R. China

Reply to R4.4: The sentence has been rephrased to accomplish other reviewers’ suggestions and the country are no longer mentioned (lines 655-666).

Round 2

Reviewer 2 Report

  1. For my question 2 “Sequences analysis was carried out with the purified amplification bands. How did the authors exclude the possibility of PCR errors and contamination?”, i think the authors’ response is inadequate. How did the authors eliminate errors in the sequencing process.
  2. Inconsistent reference format. eg. reference 34, 37 and 38.

Author Response

R.1 For my question 2 “Sequences analysis was carried out with the purified amplification bands. How did the authors exclude the possibility of PCR errors and contamination?”, i think the authors’ response is inadequate. How did the authors eliminate errors in the sequencing process.

Reply to R1: In order to better clarify the reply to referee, the samples were initially screened using a highly sensitive and specific real-time PCR, which is close system that prevents the carry-over contamination. Samples tested positive to the real-time PCR were submitted to conventional PCR with primers specific for CaCVs to obtain amplicons useful for subsequent sequencing. Contaminations were avoided through the application of rigorous laboratory procedures and good laboratory practices (i.e., handling samples separately from positive controls and using negative controls and extraction controls during extraction and real-time/conventional PCR). The amplicons with the correct size were purified using specific enzymes (Exo1 and FastaP) or gel purified to remove primer dimers and/or aspecific bands (incorrect size). Purified samples were subsequently sent to an external laboratory (Eurofins Genomics) to perform sanger sequencing. The sanger sequences obtained were subjected to sequence analyses using Geneious software version 10.2.4 (Biomatters, Auckland, New Zealand). Hight quality sequences (>95%) were subjected to BLAST N and FASTA Nucleotide search in order to retrieve the best hit in the NCBI and EBI sequence databases, respectively. This detailed information was added to the text of the newly revised manuscript (lines 130-145) .

R.2 Inconsistent reference format. eg. reference 34, 37 and 38.

 Reply to R2: The references have been now provided in the correct format.